# Moderating Effects of Organizational Climate on the Relationship between Emotional Labor and Burnout among Korean Firefighters

**DOI:** 10.3390/ijerph18030914

**Published:** 2021-01-21

**Authors:** Da-Yee Jeung, Sei-Jin Chang

**Affiliations:** 1Department of Dental Hygiene, Hanyang Womans University, Seoul 04763, Korea; cocojdy@hywoman.ac.kr; 2Department of Preventive Medicine, Yonsei University Wonju College of Medicine, Wonju 26426, Korea; 3Institute of Environmental & Occupational Medicine, Yonsei University Wonju College of Medicine, Wonju 26426, Korea

**Keywords:** burnout, climate, cross-sectional studies, emotions, firefighters

## Abstract

This study examined the association of emotional labor and organizational climate with burnout and elucidated the moderating effect of organizational climate on the relationship between emotional labor and burnout among 18,936 Korean firefighters (male: 17,790, 93.9%, female: 1146, 6.1%). To examine the effects of organizational climate on the relationships between five sub-scales of emotional labor and burnout, four groups were created using various combinations of emotional labor (“normal” vs. “risk”) and organizational climate (“good” vs. “bad”): (1) “normal” and “good” (Group I), (2) “normal” and “bad” (Group II), (3) “risk” and “good” (Group III), and (4) “risk” and “bad” (Group IV). A hierarchical multiple linear regression analysis indicated that firefighters’ burnout was significantly higher in the group with “bad” than “good” organizational climate and was significantly higher among people with “risk” than “normal” emotional labor. Combined effects of organizational climate with emotional labor on burnout were observed in all five sub-scales. Groups II, III, and IV were more likely to experience burnout than Group I (trend *p* < 0.001). Additionally, the moderating effects of organizational climate on the relationship between the five sub-scales of emotional labor and burnout were observed, except for factor 5. These results emphasize the importance of stress management to alleviate burnout caused by emotional labor at the organizational level and coping strategies to reinforce the personal potentiality suitable to organizational norms at the individual level.

## 1. Introduction

Firefighters are exposed to several kinds of stress. Although they are required to perform rigorous training to become certified and maintain continuing education, experiencing critical and hazardous events in the lines of duty can affect all areas of life. Critical events contribute to a stressful impact sufficient to overwhelm an individual’s sense of control, connection, and meaning in his/her life [1].

Although firefighters are highly trained and those who are selected to enter into this profession likely have the personal characteristics to cope with the stressors, prolonged exposure to high-stress situations and hazardous working environments may indeed impact both physical and emotional well-being [2]. Many male firefighters who suffered from traumatic events may turn to PTSD symptoms [3] and substance abuse, including drugs or alcohol [4] to deal with periods of unusually high job-related stress. Others may experience overeating, excessive weight gain, or failure to maintain a healthy lifestyle required for mental and physical health. Additional health risk factors are smoking and tobacco use [2]. These related health problems may be associated with detrimental effects on marital relationships [5]. Furthermore, if firefighters do not acquire effective coping mechanisms to deal with the stress, this deficit in skills can result in negative effects on their health and wellness such as suicidal ideation [6].

Firefighters clearly take pride in their work, perhaps reflecting the positive and confident image of these professionals [2]. In general, firefighters appear to have a sense of calling to this line of work and hold their duties in high honor, but most of them may hide behind this calling instead of developing effective coping mechanisms. Thus, firefighters might hide their true emotions and feelings related to the tough situations and events they encounter. This unique and strong culture of denial and suppression of feelings [5] may increase the risk of job dissatisfaction as firefighters realize that they are unable to cope with the emotional pressures they face while on shift. In these respects, firefighters have been considered to be workers who perform emotional labor. This unique and intense climate of firefighters as well as the high emotional demanding or suppression of emotions may play a significant role in the development of burnout.

Because of the unique nature of the work, firefighters provide an interesting and meaningful occupational group to investigate associations with burnout. However, few studies have examined the stresses that firefighters encounter on the job and the specific effects resulting from stress and burnout. Because firefighting is a mentally and physically demanding job, it can be assumed that firefighters are indeed more prone to experiencing burned-out.

Burnout and stress can have negative impacts on individuals, their families, coworkers, organizations, and even nations as a whole. Maslach and Nelson [7] demonstrated the consequences of burnout: the costs to individuals, enterprises, and society are tremendous. These costs include absenteeism, high employee turnover, new hiring, training, and re-training. When employees are error-prone, the quality of the product or service they provide suffers. Burnout manifestations created by work are also associated with a decreased quality of life at home [8]. Burnout and stress can have negative consequences that affect not only the individual experiencing these feelings but their coworkers and the larger organization as well. Stress and burnout have been related to increased employee turnover, increased intention to leave, reduced levels of performance, and both physical and psychological illness [9,10]. This creates an unpleasant work environment for the burned-out individual and places greater responsibility on coworkers [8].

A growing body of literature has identified and emphasized the moderating factors that amplify or buffer the negative results of emotional labor. Some studies have identified organizational culture and climate as moderating factors in the relationship between emotional labor and its adverse outcomes [11,12]. The concept of organizational climate has been widely discussed. Jorde–Bloom defined organizational climate as “the awareness of the organization members of the status and situation of the organization” [13]. Organizational climate may play a crucial role in relation to members’ behavior, levels of motivation, and organizational commitment [14,15]. A cooperative and friendly organizational climate among members may alleviate negative conditions such as emotional disharmony or dissonance. In turn, fewer negative conditions may lead to a decrease in the risk of burnout. Although firefighters cannot avoid emotional labor, a positive organizational climate could be achieved through organizations’ internal efforts to reduce job stress, and to weaken negative outcomes. For example, Ryu et al. reported that the combined effect of organizational climate on the relationship between emotional labor and turnover intention among firefighters [16].

The purposes of this study were to examine the effects of emotional labor and organizational climate on burnout and to identify and elucidate the moderating effects of organizational climate on the relationship between emotional labor and burnout among Korean firefighters.

## 2. Materials and Methods

### 2.1. Study Participants

Data were obtained from the study Firefighters Research: Enhancement of Safety and Health (FRESH), funded by the National Fire Agency in Korea. Of 42,738 registered Korean firefighters, 18,936 firefighters (male: 17,790 (93.9%), female: 1146 (6.1%)) whose main duties were “firefighting (*n* = 8859),” “rescue (*n* = 1943),” “emergency medical aid (*n* = 5129),” “administration (1957),” and “others (*n* = 1048) were voluntarily included in this study (response rate: 43.3%). The sample of this study was proportional to region and main duty of the total Korean firefighters, and it might be considered as a representative and nationwide sample.

### 2.2. Procedures

We collected the data using an online (http://fresh.re.kr/login.php) self-reported questionnaire from 20 August 2016 to 31 January 2017, which comprised the following: (1) general characteristics (7 items), (2) job characteristics (3 items), (3) emotional labor (26 items), (4) organizational climate (5 items), and (5) burn out (5 items).

### 2.3. Instruments

Emotional labor was measured using the Korean Emotional Labor Scale, which consists of five sub-scales [17]: (1) emotional demands and regulation (5 items), (2) overload and conflict in customer service (3 items), (3) emotional disharmony and hurt (6 items), (4) organizational surveillance and monitoring (3 items), and (5) lack of a supportive and protective system in the organization (7 items). Each sub-scale was rated on a scale of 1 (disagree completely), 2 (disagree), 3 (agree), and 4 (agree completely). Next, the five sub-scales of emotional labor were divided into “normal” and “risk” groups according to the guidelines of the Korea Emotional Labor Scale [17]. The reliability coefficients (Cronbach’s α) for the five sub-scales of emotional labor used in this study ranged from 0.770 to 0.929.

Organizational climate was measured using a five-item scale from the FRESH study, as described below:(1)There is good harmony between members of the organization.(2)There are many disputes or arguments about how to work among the members.(3)There is no integration between members.(4)The relationships between my department and others are good.(5)There is good cooperation between the departments I belong to and other departments.

Items (1), (4), and (5) were rated using a 4-point scale: 1 = “disagree completely,” 2 = “disagree,” 3 = “agree”, and 4 = “agree completely”. Items (2) and (3) were rated through reverse scoring. Organizational climate was calculated by summing the five items, with total scores ranging from 5 to 20. The reliability coefficients (Cronbach’s α) for organizational climate was 0.847.

Burnout was assessed using the 5-item scale developed by Maslach and Jackson [18]. Each item was measured on a 4-point Likert scale ranging from 1 (“not at all”) to 4 (“very often”). Higher total scores on this measure indicate a higher degree of burnout. The following is an example item: “I feel exhausted because of my work”. In the present study, Cronbach’s α was 0.889.

### 2.4. Study Design and Research Hypothesis

This study was cross-sectional, and designed to examine the moderating effects of organizational effects on the relationship between emotional labor and burnout in firefighters (Figure 1). To test this objective, we proposed the following hypotheses. First, emotional labor is positively associated with burnout (H1). Second, a cooperative organizational climate is related with burnout (H2). Third, organizational climate reduces the negative effects of emotional labor on burnout (H3).

### 2.5. Statistical Analysis

To compare burnout according to socio-demographic and job characteristics, we conducted a t-test or ANOVA. To examine the effects of organizational climate on the association between emotional labor and burnout, three statistical methods were performed. First, we analyzed the differences of burnout by emotional labor (“normal” vs. “risk”) through the sub-group analysis stratified organizational climate (“good” vs. “bad”). Second, we examined the differences in burnout according to the various combinations of the five sub-scales of emotional labor and organizational climate. To do this, four groups were created using various combinations of emotional labor (“normal” vs. “risk”) and organizational climate (“good” vs. “bad”): (1) “normal” and “good” (Group I), (2) “risk” and “good” (Group II), (3) “normal” and “bad” (Group III), and (4) “risk” and “bad” (Group IV). Third, a hierarchical multiple linear regression analysis was conducted to confirm the moderating effects of organizational climate on the relationship between emotional labor and burnout. SPSS/PC (Version 23) was used for the analyses, and *p* < 0.05 was considered significant.

### 2.6. Ethical Consideration

This study was approved by the Institutional Review Board of Yonsei University for the protection of the rights and privacy of participants (Approval No. CR318335), and was conducted in full accordance with the World Medical Association Declaration of Helsinki. All participants were given information about the study and were asked to sign a consent form prior to their participation, and their responses were anonymously guaranteed.

## 3. Results

### 3.1. Differences in Burnout According to Socio-Demographic and Job Characteristics

Among the participants in this study, 17,790 (93.9%) were male, 8274 (43.7%) were under the age of 40 (mean: 42.2, standard deviation: 8.8), 10,331 (54.6%) graduated from a 4-year college, 9156 (48.4) worked for less than 10 years, and 16,551 (87.4%) were shift workers. The most frequent shift work was three shifts (*n* = 14,791: 78.1%). The main duties of the participants were firefighting (46.8%), emergency medical aid (27.1%), administration or education (10.3%), rescue (10.3%), and others (4.4%).

Burnout was associated with gender (*p* < 0.001), age (*p* < 0.001), education (*p* < 0.001), shift work (*p* < 0.001), main duty (*p* < 0.001), and working period (*p* < 0.001) (Table 1).

### 3.2. Differences of Burnout According to the Sub-Scales of Emotional Labor and Organizational Climate

To examine the differences in burnout according to the sub-scales of emotional labor and organizational climate, t-tests were conducted by categorizing each factor into two groups (normal vs. risk, good vs. bad) using median values. As shown in Table 2, all sub-scales of emotional labor and organizational climate were associated with burnout (*p* < 0.001). Firefighters with higher levels of emotional labor were more likely to experience burnout compared to those with normal levels of emotional labor. In addition, burnout was significantly higher among firefighters in a bad organizational climate than in those in a good organizational climate.

### 3.3. Combined Effects of Organizational Climate with Emotional Labor on Burnout

We analyzed the relationships of the various combinations of the five sub-scales of emotional labor and organizational climate with burnout after adjustment for control variables. In the relationship between emotional labor and burnout, the combined effects of organizational climate were observed in all five sub-scales. Compared to Group I, Group II, Group III, and Group IV were more likely to experience burnout (*p* for trend <0.001). Emotional labor resulted in increased risk of burnout among firefighters who were working in a “bad” organizational climate compared to those in a “good” organizational climate. We discovered that the levels of burnout of firefighters were highest among Group IV (“risk” emotional labor and “bad” organizational climate) and lowest among Group I (“normal” emotional labor and “good” organizational climate) (Table 3).

### 3.4. Correlation Coefficients between Emotional Labor, Organizational Climate, and Burnout

To assess whether the sub-scales of emotional labor and organizational climate are related to burnout, we performed correlation analysis. As shown in Table 4, while five sub-scales of emotional labor (“emotional demand and regulation” (r = 0.353, *p* < 0.01), “overload and conflict in customer service” (r = 0.354, *p* < 0.01), “emotional disharmony and hurt” (r = 0.447, *p* < 0.01), “organizational surveillance and monitoring” (r = 0.370, *p* < 0.01), and “lack of a supportive and protective system in the organization” (r = 0.242, *p* < 0.01)) were positively correlated with burnout, “organizational climate” (r = −0.308, *p* < 0.01) was negatively correlated with burnout.

### 3.5. Moderating Effects of Organizational Climate on the Relationship between Emotional Labor on Burnout

A hierarchical multiple linear regression analysis was performed to identify the moderating effects of organizational climate on the relationship between each sub-scale of emotional labor on burnout after adjustment for control variables (Table 5). Model 1 included the five sub-scales of emotional labor separately. In Model 2, we added “organizational climate”. Finally, five sub-scales of emotional labor x occupational climate (interaction terms) were added in Model 3. As a result, the fits of the three models were statistically significant. Analysis using the variance inflation factor (VIF) indicated no issues related to multi-collinearity among independent variables.

In Model 1, the five sub-factors of emotional labor, “emotional demands and regulation” (b = 0.143, *p* < 0.001, R^2^ = 0.172), “overload and conflict in customer service” (b = 0.120, *p* < 0.001, R^2^ = 0.175), “emotional disharmony and hurt” (b = 0.160, *p* < 0.001, R^2^ = 0.247), “organizational surveillance and monitoring” (b = 0.140, *p* < 0.001, R^2^ = 0.190), and “lack of a supportive and protective system in the organization” (b = 0.123, *p* < 0.001, R^2^ = 0.123) were associated with burnout. In Model 2, five sub-scales of emotional labor were still associated with burnout (“emotional demands and regulation” (b = 0.126, *p* < 0.001, R^2^ = 0.240), “overload and conflict in customer service” (b = 0.102, *p* < 0.001, R^2^ = 0.234), “emotional disharmony and hurt” (b = 0.138, *p* < 0.001, R^2^ = 0.280), “organizational surveillance and monitoring” (b = 0.111, *p* < 0.001, R^2^ = 0.229), and “lack of a supportive and protective system in the organization” (b = 0.070, *p* < 0.001, R^2^ = 0.176)), and organizational climate was negatively related to burnout. The total variance of burnout (R^2^) displayed a significant increase in all five sub-scales. In Model 3, we added the interaction terms of the five sub-scales of emotional labor x organizational climate in the model to examine the moderating effects of organizational climate on the relationship between emotional labor and burnout. The four interaction terms with the exception of “lack of a supportive and protective system in the organization” were negatively associated with burnout. These results suggest that a positive organizational climate plays a significant role in reducing the negative effects of emotional labor on burnout.

## 4. Discussion

Firefighters are well known as workers who perform emotional labor. They must manage traumatic scenarios that require emotional labor, a psychosocial stressor known to be related to burnout in human service work. Because of the unique nature of the work, firefighters pose an interesting and meaningful occupational group to investigate in relation to burnout. The unique climate of firefighting puts these workers at risk of experiencing burnout. However, few studies have examined the job-related stresses that firefighters encounter and the specific effects resulting from stress and burnout. Not only are the consequences of burnout serious to the individual’s mental/physical/emotional state, but also, especially in firefighting, to the safety of that individual, their co-workers, and the surrounding community that a fire may threaten. Firefighters embody distinct qualities that set them apart from traditional forms of employment. In short, firefighters have very specific duties they are employed to conduct. Consequently, most of their work tasks are dissimilar to those of white-collar workers or typical business-related jobs.

Burnout is not intuitively related to firefighting. Therefore, gaps remain in our knowledge regarding the link between firefighting and burnout. According to Gossett [19], the rapid growth in the number of temporary workers and its significance for our overall economy have led researchers to examine the impact of temporary work arrangements on our current theories of organization and communication. Consequently, several questions arose: Does burnout occur in these types of jobs? What factors contribute to burnout in an intense but temporary position? What do firefighters do differently that helps them mitigate burnout? Due to the emotional and physical stresses of firefighting, the extreme conditions and hazardous work environment, and the inherent nature of firefighter shift work, burnout occurs despite the temporary nature of the job. Burnout is considered a felt state. Therefore, the discursive construction of those feelings/emotions expressed through firefighter communication can be linked to their experience of burnout [20].

The present study revealed that high levels of emotional labor were associated with burnout among firefighters, while a friendly or cooperative organizational could reduce it. A growing body of literature has suggested that the association between emotional labor and burnout may not be a direct causal relationship. Prolonged exposure to high levels of emotional labor causes negative and remarkable work adjustment [21] and job satisfaction [22], which in turn affects burnout and turnover intention. Nixon et al. [23] also supported the theory that emotional labor in the workplace aggravates psychological stress and causes adverse consequences, such as turnover. In addition, Zapf [24] revealed that emotional labor, when combined with organizational problems, is associated with burnout and may be positively related to turnover. Burnout and its effects can manifest in numerous ways, depending on individuals and their specific occupation, as well as the social dynamics present in the work environment. Miller [25] argued that negative psychological, physiological, and organizational outcomes are all associated with burnout. Negative consequences are suffered not only on the individual level but the organizational and societal level as well. Social withdrawal, depression, lack of motivation, and lack of personal investment are merely a few expressions of the burnout phenomenon as observed within an individual. A recent work by Back et al. [26] supports this result by asserting that burnout mediates in the association between emotional labor and turnover intention in clinical nurses. For a burned-out individual, symptoms may also include a diminished sense of personal accomplishment, decreased efficiency at work, increased feelings of cynicism and hostility and reduced professional commitment [7,27,28,29].

Some mechanisms provide theoretical explanations for how emotional labor leads to burnout. Jeung et al. [30] demonstrated the reasoning for the effects of emotional labor on burnout in two ways. First, when personal resources are threatened or lost, these losses result in anxiety and distress in the individual, thereby increasing physiological arousal and adverse health problems [31]. Previous research has documented that employees are likely to respond to angry or rude customers by suppressing or faking emotional expressions [32]. Such chronic and prolonged self-regulatory efforts may lead to a loss of resources, for several reasons. First, the inauthenticity of faking expressions, or surface acting, decreases one’s self-worth and self-efficacy [33]. Second, suppressing emotions is expected to require energy resources, as demonstrated by increased physiological arousal, higher levels of glucose, and reduced motivation [34]. Consequently, chronic and frequent exposure to stress due to excessive emotional demands may activate the stress system, including the hypothalamic–pituitary–adrenal (HPA) axis and the sympathetic nervous system. Third, suppressing emotions results not in directly changing those feelings, but in fewer social interactions with others [35], which in turn reduces social resources.

A second explanation for the mechanisms of the causal relationship between emotional labor and burnout has focused on emotional acting: surface acting. Surface acting is more likely to cause emotional exhaustion due to the effort required to hide or suppress negative emotions [9]. The use of surface acting has consistently been revealed to produce emotional exhaustion that threatens well-being [36]. Previous research indicates that surface acting is likely to deplete energy, as it involves prolonged internal strain between one’s displayed and true feelings, which consequently causes emotional dissonance. Recent research suggests that accepting external influences and acting against one’s internal emotions are significantly related to anxiety, stress, and aggravated subjective and psychological wellness [37]. The long-lasting experience of emotional exhaustion is more likely to increase the risk of high levels of psychological effort, thereby leading to the loss of resources [38,39], and finally resulting in burnout. Surface acting is significantly related to negative affect and work withdrawal [40]. Overall, a growing body of research has documented that faking or suppressing one’s true emotions is related to stress, resource depletion, and burnout [34].

As a modifier of the adverse outcomes of emotional labor, organizational climate can play a crucial role in improving employers’ behavior and levels of motivation and organizational commitment [14,15]. A cooperative and friendly organizational climate among members may weaken the negative factors such as emotional suppression, regulation, and emotional dissonance. In this study, we identified a positive association between good organizational climate and lower levels of burnout.

In the present study, we observed a combined effect of organizational climate with emotional labor on burnout. When an organizational climate is supportive or cooperative, even if emotional labor is high, burnout decreases. In other words, an authentic friendly organizational climate among employers mitigates against the transformation of emotional labor to burnout. This supports Cheng et al.’s finding that a positive organizational climate mediates the consequences of burnout and turnover intention [41]. However, a negative or non-supportive organizational climate amplifies the impact of emotional labor on burnout. This supports the observation that a friendly and supportive organizational climate within an organization has a strong, positive effect on job satisfaction [42]. As a result, a good organizational climate for both employers and organizations may play a role in reducing the negative aspects of emotional labor on burnout among firefighters.

While previous studies on emotional labor have focused on negative outcomes or mental health problems, such as job stress, job satisfaction, and burnout, this study attempted to evaluate the moderating effects of organizational climate on the relationship between emotional labor and burnout. Previous studies have been conducted on mediating variables such as burnout, job stress, and emotional dissonance [43,44]. In the present study, it was confirmed that organizational climate plays an important role in reducing burnout. Thus, it is crucial to build a favorable organizational climate for a firefighting organization to mitigate burnout among firefighters.

In accordance with building a positive organizational climate, coping strategies to promote mental health at the individual level need to be provided. Previous studies point to the prevalence of posttraumatic stress disorder in military personnel, but PTSD is also common among firefighters. Cicognani et al. [1] identified both negative and positive outcomes of emergency work. They considered coping strategies, self-efficacy, and collective efficacy in investigating emergency workers’ quality of life and related eight psychosocial facets. Results suggested that burnout was associated with inadequate coping strategies, such as distraction and self-criticism, and that the sense of community and collective efficacy were slightly, while self-efficacy was strongly, linked to well-being outcomes [1].

This study had a few limitations. First, it analyzed a nationwide sample of firefighters, but we cannot rule out the possibility of regional differences. Second, although this cross-sectional study revealed a significant relationship between emotional labor and burnout as well as moderating effects of organizational climate on this relationship, it is difficult to conclude a causal relationship. These limitations need to be considered when examining the causal relevance of organizational climate through prospective cohort studies in the future.

Despite these limitations, this study is meaningful in that it identifies the combined effects of organizational climate with emotional labor on burnout. Because firefighters cannot avoid emotional dissonance at work, they need to modify the organizational climate in their work environment. Improving the organizational climate will promote well-being among firefighters.

## 5. Conclusions

The results of this study suggest that emotional labor, as a newly emergent job stressor, leads to burnout among firefighters. These results also emphasize the importance of stress management programs to reduce or alleviate the negative outcomes caused by emotional labor at the organizational level, and coping strategies to reinforce the personal potentiality suitable to organizational norms and work settings at the individual level. In addition, the guidelines and training for minimizing the negative effects of emotional labor among firefighters, such as burnout and depression, should be developed. Coping strategies to strengthen self-regulatory variables and an adaptive personality suitable to organizational norms are needed at the individual level. It is also necessary to construct a protective and supportive management system at the organizational level. Furthermore, legislation for the prevention of negative outcomes caused by emotional labor and healthy consumerism are needed at the state level. The results of this study contribute to the understanding of how emotional labor and organizational climate can affect burnout in firefighters.

## Figures and Tables

**Figure 1 ijerph-18-00914-f001:**
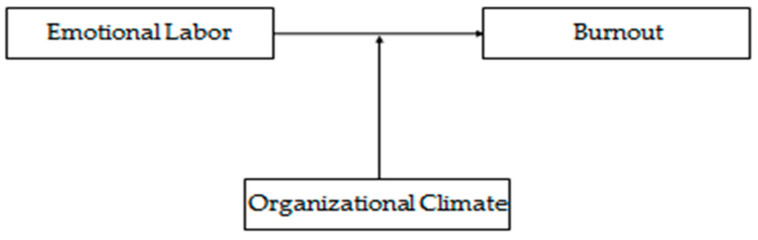
Research hypothesis of this study.

**Table 1 ijerph-18-00914-t001:** Mean values and standard deviations of burnout according to the socio-demographic and job characteristics.

Variables		Mean ± S.D.	*p* Value
Gender			<0.001
	Male (*n* = 17,790)	14.51 ± 7.25	
	Female (*n* = 1146)	18.93 ± 8.11	
Age			<0.001
	20–29 *n* = 1191)	13.93 ± 7.90	
	30–39 (*n* = 7083)	15.72 ± 7.78	
	40–49 (*n* = 6021)	15.11 ± 7.18	
	50- (*n* = 4641)	13.13 ± 6.50	
Education			<0.001
	Less than or equal to junior college graduate (*n* = 10,331)	14.39 ± 7.30	
	4 years-college graduate (*n* = 8309)	15.23 ± 7.41	
	More than graduate school (*n* = 296)	15.67 ± 8.23	
Shift work			<0.001
	No shift (*n* = 2385)	16.47 ± 8.42	
	24-h shift (*n* = 580)	14.54 ± 7.39	
	Two shifts (*n* = 342)	14.56 ± 7.45	
	Three shifts (*n* = 14,791)	14.59 ± 7.16	
	Other (*n* = 838)	13.55 ± 7.18	
Main duty			<0.001
	Firefighter (*n* = 8859)	13.79 ± 6.74	
	Rescue (*n* = 1943)	12.42 ± 6.22	
	Emergency medical aid (*n* = 5129)	16.22 ± 7.66	
	Administration (*n* = 1957)	16.77 ± 8.40	
	Other (*n* = 1048)	16.74 ± 8.24	
Working period			<0.001
	Less than 5 years (*n* = 4911)	14.96 ± 7.90	
	5–9 years (*n* = 4245)	15.86 ± 7.58	
	10–14 years (*n* = 2966)	15.53 ± 7.23	
	More than 15 years (*n* = 6814)	13.64 ± 6.73	

(*n* = 18,936).

**Table 2 ijerph-18-00914-t002:** Mean values and standard deviations (SD) of burnout according to the five sub-scales of emotional labor.

Variables		Mean ± SD	*p*
Emotional Labor			
EDR	Normal (*n* = 16,703)	14.15 ± 7.00	<0.001
	Risk (*n* = 2233)	19.49 ± 8.36	
OCCS	Normal (*n* = 17,295)	14.24 ± 7.03	<0.001
	Risk (*n* = 1641)	20.46 ± 8.49	
EDH	Normal (*n* = 17,056)	14.02 ± 6.89	<0.001
	Risk (*n* = 1880)	21.66 ± 8.03	
OSM	Normal (*n* = 15,603)	13.80 ± 6.91	<0.001
	Risk (*n* = 3333)	19.35 ± 7.78	
LSPSO	Normal (*n* = 12,098)	13.61 ± 6.93	<0.001
	Risk (*n* = 6838)	16.85 ± 7.68	
Organizational climate	Good (*n* = 14,023)	13.58 ± 6.86	<0.001
	Bad (*n* = 4913)	18.19 ± 7.72	

EDR: Emotional demands and regulation, OCCS: Overload and conflict in customer service, EDH: Emotional disharmony and hurt, OSM: Organizational surveillance and monitoring, LSPSO: Lack of a supportive and protective system in the organization, OC: Organizational climate.

**Table 3 ijerph-18-00914-t003:** Mean values and standard deviations (SD) of burnout according to the four groups of the combinations of emotional labor with organizational climate.

Emotional Labor & Organizational Climate	*n* (%)	Mean ± SD	*p* for Trend
EDR ‘Normal’ & OC ‘Good’	12,621(66.65)	13.11 ± 6.55	<0.001
EDR ‘Risk’ & OC ‘Good’	4082(21.56)	17.36 ± 7.35	
EDR ‘Normal’ & OC ‘Bad’	1402(7.40)	17.85 ± 8.03	
EDR ‘Risk’ & OC ‘Bad’	831(4.39)	22.24 ± 8.20	
OCCS ‘Normal’ & OC ‘Good’	13,009(68.70)	13.19 ± 6.59	<0.001
OCCS ‘Risk’ & OC ‘Good’	4286(22.63)	17.43 ± 7.36	
OCCS ‘Normal’ & OC ‘Bad’	1014(5.35)	18.64 ± 8.21	
OCCS ‘Risk’ & OC ‘Bad’	627(3.31)	23.41 ± 8.11	
EDH ‘Normal’ & OC ‘Good’	12,951(68.39)	13.03 ± 6.47	<0.001
EDH ‘Risk’ & OC ‘Good’	4105(21.68)	17.13 ± 7.24	
EDH ‘Normal’ & OC ‘Bad’	1072(5.66)	20.21 ± 7.88	
EDH ‘Risk’ & OC ‘Bad’	808(3.31)	23.59 ± 7.83	
OSM ‘Normal’ & OC ‘Good’	12,274(64.82)	12.95 ± 6.52	<0.001
OSM ‘Risk’ & OC ‘Good’	3329(17.58)	16.93 ± 7.39	
OSM ‘Normal’ & OC ‘Bad’	1749(9.24)	18.00 ± 7.56	
OSM ‘Risk’ & OC ‘Bad’	1584(8.37)	20.85 ± 7.74	
LSPSO ‘Normal’ & OC ‘Good’	10,115(53.42)	12.87 ± 6.60	<0.001
LSPSO ‘Risk’ & OC ‘Good’	1983(10.47)	17.39 ± 7.34	
LSPSO ‘Normal’ & OC ‘Bad’	3908(20.64)	15.44 ± 7.17	
LSPSO ‘Risk’ & OC ‘Bad’	2930(15.47)	18.73 ± 7.93	

EDR: Emotional demands and regulation, OCCS: Overload and conflict in customer service. EDH: Emotional disharmony and hurt, OSM: Organizational surveillance and monitoring. LSPSO: Lack of a supportive and protective system in the organization. OC: Organizational climate.

**Table 4 ijerph-18-00914-t004:** Correlation coefficients among burnout, five sub-scales of emotional labor, and organizational climate.

	EDR	OCCS	EDH	OSM	LSPSO	OC
Burnout	0.353 **	0.354 **	0.447 **	0.370 **	0.242 **	−0.308 **
EDR	1	0.607 **	0.616 **	0.495 **	0.205 **	−0.147 **
OCCS		1	0.737 **	0.589 **	0.190 **	−0.197 **
EDH			1	0.681 **	0.263 **	−0.317 **
OSM				1	0.260 **	−0.340 **
LSPSO					1	−0.418 **

EDR: Emotional demands and regulation, OCCS: Overload and conflict in customer service. EDH: Emotional disharmony and hurt, OSM: Organizational surveillance and monitoring. LSPSO: Lack of a supportive and protective system in the organization. OC: Organizational climate. **: *p* < 0.01.

**Table 5 ijerph-18-00914-t005:** Hierarchical multiple linear regression analysis of the five-sub-scales of emotional labor, organizational climate and interaction terms (emotional labor * organizational climate) on burnout.

	Model 1		Model 2		Model 3	
	B (t)	*p*	B (t)	*p*	B (t)	*p*
EDR	0.143 (48.48)	<0.001	0.126 (44.15)	<0.001	0.225 (13.63)	<0.001
OC			−0.981 (−41.22)	<0.001	−0.558 (−7.62)	<0.001
EDR * OC					−0.006 (−6.10)	<0.001
F	217.870	<0.001	314.335	<0.001	301.053	<0.001
Adjusted R^2^	0.172		0.240		0.241	
OCCS	0.120 (49.65)	<0.001	0.102 (42.65)	<0.001	0.184 (13.53)	<0.001
OC			−0.922 (−38.21)	<0.001	−0.623 (−11.45)	<0.001
OCCS * OC					−0.005 (−6.14)	<0.001
F	224.736	<0.001	306.156	<0.001	293.297	<0.001
Adjusted R^2^	0.175		0.234		0.236	
EDH	0.160 (67.13)	<0.001	0.138 (56.12)	<0.001	0.245 (18.97)	<0.001
OC			−0.714 (−29.63)	<0.001	−0.373 (−7.95)	<0.001
EDH * OC					−0.007 (−8.48)	<0.001
F	346.521	<0.001	389.700	<0.001	375.198	<0.001
Adjusted R^2^	0.247		0.280		0.283	
OSM	0.140 (53.36)	<0.001	0.111 (40.90)	<0.001	0.195 (14.05)	<0.001
OC			−0.782 (−31.01)	<0.001	−0.541 (−11.64)	<0.001
OSM * OC					−0.006 (−6.17)	<0.001
F	247.482	<0.001	296.978	<0.001	284.582	<0.001
Adjusted R^2^	0.190		0.229		0.230	
LSPSO	0.123 (34.60)	<0.001	0.070 (18.715)	<0.001	0.076 (4.44)	<0.001
OC			−0.931 (−34.80)	<0.001	−0.913 (−15.42)	<0.001
LSPSO * OC					0.000 (−0.349)	0.727
F	149.056	<0.001	213.973	<0.001	203.271	<0.001
Adjusted R^2^	0.123		0.176		0.176	

Model 1: Adjustment for gender, age, education, shift work, main duty, and working period. Model 2: Adjustment for Model 1 + organizational climate. Model 3: Adjustment for Model 2 + organizational climate * occupational climate (interaction term). EDR: Emotional demands and regulation, OCCS: Overload and conflict in customer service, EDH: Emotional disharmony and hurt, OSM: Organizational surveillance and monitoring, LSPSO: Lack of a supportive and protective system in the organization, OC: Organizational climate.

## Data Availability

We collected the data using an online (http://fresh.re.kr/login.php) self-reported questionnaire from 20 August 2016 to 31 January 2017.

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
