# Peer review of "Moderating Effects of Organizational Climate on the Relationship between Emotional Labor and Burnout among Korean Firefighters"

_ijerph, 2021, doi:10.3390/ijerph18030914_

Round 1

Reviewer 1 Report

It is a manuscript that describes a profession or job that has not been widely studied. For this reason, it is necessary to continue studying the burnout of the professional firefighters.

Below are a number of considerations if they may be useful.

It is recommended to introduce socio-demographic information regarding gender in the abstract. In particular, one could introduce what is the percentage of men or women and the average age and its standard deviation.
Put the key words in alphabetical order, also trying to use a key word that refers to the research design used.
It would be interesting to deal in greater depth with a general theory that describes the phenomenon and helps to design hypotheses.
A general objective appears. It would be interesting to introduce specific objectives and hypotheses.
The section on "measures" should be divided into procedure and instruments. 
The results are very well structured. It would be interesting if the sub-sections were related to hypotheses.
The discussion is very well developed. This section connects very well with the introduction and compares other studies with the results of this study. Furthermore, it describes the limitations of the study and future lines of research. 
The conclusions are also appropriate.
It is necessary to comment on where Table 1 would go. It does not appear previously.
The biliography is related to the research. In any case, it is recommended that more bibliographical references from the last five years be used. 

That said, this is a manuscript that deals with an interesting subject and on which further research should be carried out. Furthermore, it has a fairly intuitive structure and I believe that with minor modifications it could be proposed for publication. Thank you very much for your attention. 

Author Response

I would like to submit the reponse to the reviewer's comments.

Please see the attachment. Thank you very much.

Reviewer 2 Report

Thank you for sending your paper entitled “Moderating Effects of Organizational Climate on the Relationship between Emotional Labor and Burnout among Korean Firefighters” to Internacional Journal Environmental Research and Public Health. After carefully review this interesting paper, the following comments are listed for your reference: 

  1. Abstract: To increase potential citations, authors should check keywords against those recommended in the MESH Browser of Medical Subject Headings https://meshb.nlm.nih.gov/search.
  2. Methods (P2): It would be desirable to include the representativeness of the sample in the “Sample and Setting” section.What was the year of data collection?
  3. Methods (P2): What is the study design? This should be added in section “Study design”
  4. Methods (P3): I would suggest to include the ethical principles of the Declaration of Helsinki. This would be “Ethical consideration”.
  5. Methods (P3): An online self-reported questionnaire was used:

How was the consent of the participants obtained?

Was anonymity guaranteed?

Was the participation voluntary?

What online platform was used for the online questionnaire?

All these issues should be explained in the method section "data collection"

5. Results (P4-P5): Tables must be attached within the results section. Table 1 is missing in the results section.

6. Final part of the manuscript needs to add: funding and autor contributions.

7. References (P13-15): 37/41 are more than 5 years old and 28 of them are more than 10 years old.  Referencing recent and relevant literature would make the paper more robust, especially in the introduction and discussion sections. If this is not the case (there is no recent literature), it should be noted appropriately in each section and limitation section.

8. References (P13-15): References 13, 14,15,21,36,39 are miscited. I recommend checking the citations according to the journal's standards. In addition, the DOI must be added in each bibliographic reference.

9. I recommend you to improve the methods section and results to make it clearer to the reader. This article has serious flaws and research not conducted correctly. 

Author Response

(The authors gave the same response as above.)

Round 2

Reviewer 2 Report

Thank you for asking me to review this manuscript. I accept this manuscript in present form.